# A comparative analysis of objectively assessed physical activity levels in kindergarten and home among children aged 5 to 6

Jarosław Herbert[1]*, Piotr Matłosz[1], Wojciech Ratkowski[2], Justyna Wyszyńska[3]

**1** Faculty of Physical Culture Sciences, Medical College, University of Rzeszów, Rzeszów, Poland,
**2** Track and Field Department, Gdansk University of Physical Education and Sport, Gdansk, Poland,
**3** Faculty of Health Sciences and Psychology, Medical College, University of Rzeszów, Rzeszów, Poland

\* jherbert@ur.edu.pl

## Abstract

Children's physical activity (PA) is positively associated with a wide range of developmental and health outcomes. This study compared individual levels of PA in children aged 5–6 years across two settings: kindergarten and home. A total of 522 children (51.9% girls) participated. PA was objectively measured using ActiGraph GT3X-BT accelerometers, and selected socioeconomic indicators (SSI) were parent-reported. Children accumulated significantly more light physical activity (PA) in kindergarten than at home, whereas moderate, vigorous, and total moderate-to-vigorous physical activity (MVPA) were significantly higher at home than in kindergarten (all $p < 0.001$). Boys showed consistently higher PA levels than girls in both environments. Notably, median MVPA in boys was 22.7 min at home and 17.3 min in kindergarten, compared to 19.0 and 14.0 min in girls, respectively. Median daily step counts in the total sample were also significantly higher at home (3315.5) than in kindergarten (3111.0). Significant associations were observed between selected SSI and PA. Lower parental education levels were associated with higher step counts, light (father's education) and moderate PA (mother's education) in kindergarten. Children from families with less favorable financial conditions also had higher MVPA in kindergarten. These findings underscore the importance of the home environment and suggest that certain aspects of socioeconomic disadvantage may be linked to higher PA levels in early childhood.

## Introduction

Physical activity (PA) plays a fundamental role in early childhood development. Adequate levels of PA in preschool children are associated with better cognitive functioning, learning outcomes, and psychosocial well-being [1]. Moreover, regular PA supports healthy somatic development by preventing obesity [2], improving

**Data availability statement:** All files are available from the database in the Zenodo, https://doi.org/10.5281/zenodo.15974941.

**Funding:** The author(s) received no specific funding for this work.

**Competing interests:** The authors have declared that no competing interests exist.

cardiovascular function [3], enhancing motor coordination, and promoting favorable body composition and bone health [4,5].

International guidelines consistently emphasize the need for sufficient daily movement in early childhood. Children under the age of five should accumulate at least 180 minutes of PA per day at any intensity, while children in the second cycle of preschool education are recommended to engage in a minimum of 60 minutes of moderate-to-vigorous physical activity (MVPA) daily [6,7]. Daily step counts of approximately 10,000–14,000 steps have also been proposed as a practical indicator of sufficient PA in this age group [8]. Despite these recommendations, many preschool populations fail to meet PA guidelines, and sedentary behavior is increasingly prevalent [9–12].

Evidence from international studies highlights substantial cross-country differences in PA levels among kindergarteners. While nearly 60% of Norwegian children meet PA recommendation [13], substantially lower compliance has been observed in Belgium (10.1%) [14] and Australia (14.9%) [15]. In the United States, approximately 41.6% of preschool children achieve recommended PA levels [16]. Systematic reviews confirm that globally, only a minority of preschool children meet 24-hour movement guidelines, particularly with respect to MVPA [17]. These findings underline the need to examine contextual determinants of PA across different settings.

Synthesized evidence from accelerometer-based studies consistently indicates that preschool children spend only a small proportion of their day engaged in MVPA. Meta-analytic and large observational studies show that MVPA typically accounts for approximately 3–6% of daily time in kindergarten settings, with similar or only slightly higher levels observed at home [18–20]. Although some studies report higher estimates, particularly when longer monitoring periods or different analytical approaches are used, overall findings suggest that MVPA constitutes a relatively limited share of preschool children's daily movement behavior [21,22].

Both the home and kindergarten environments are key determinants of PA in early childhood. Previous research indicates that family routines, parental behaviors, and access to recreational spaces influence children's activity levels [23–25]. In kindergarten settings, policy-level and structural factors, such as indoor space availability, group size, staff training, and daily scheduling, can substantially restrict or facilitate opportunities for both spontaneous and structured PA. Evidence shows that limited indoor space and reduced access to outdoor play are associated with lower MVPA levels, even when national PA guidelines are in place [26,27].

Beyond organizational constraints, specific environmental characteristics of kindergartens further shape children's activity patterns. Kindergarten features such as time allocated for outdoor play, availability of equipment, and overall organization of daily routines influence opportunities for movement during the day [28–30].

Recent systematic reviews and meta-analytic evidence indicate that outdoor environmental features, including open spaces, natural elements, fixed and mobile equipment, and opportunities for ball and object play, are positively associated with children's MVPA, whereas sedentary-oriented environments are linked to lower activity levels [31,32]. Consistently, international evidence suggests that cross-country

differences in kindergarten PA levels are shaped more by contextual and institutional practices than by national guidelines alone [33].

Accurate assessment of PA in preschool children remains methodologically challenging. Accelerometry is widely used for the objective assessment of PA in young children; however, estimates of activity intensity are highly sensitive to methodological decisions, including device wear-time compliance, epoch length, cut-point selection, and data processing. Recent evidence highlights that these issues substantially affect estimates of MVPA and sedentary behavior in preschool populations. In particular, comparative analyses show that applying different accelerometer cut-points and epoch lengths to the same dataset can result in markedly different classifications of time spent in MVPA and sedentary behavior, thereby limiting comparability across studies. Given the sporadic and intermittent nature of preschool children's movement patterns, shorter epoch lengths have been shown to better capture higher-intensity activity, whereas longer epochs tend to underestimate it [34,35].

Recent systematic reviews further report substantial heterogeneity in accelerometer protocols used in preschool research, including differences in device models, cut-points, and data processing strategies, which hampers direct comparison of PA outcomes. This methodological variability is particularly evident in studies conducted in kindergarten settings, where differences in daily routines and activity contexts further interact with measurement choices [36].

In Poland, evidence on objectively measured PA in preschool children is scarce. Most previous studies relied on subjective methods, such as questionnaires and surveys [37], limiting comparability with international research. This gap is particularly important given that Polish children spend an average of 35 hours per week in kindergarten, more than their peers in North America and comparable to the longest childcare attendance in Europe [38–40]. Socioeconomic conditions further influence children's PA opportunities, as families with higher socioeconomic status have better access to facilities and resources that support active lifestyles [41,42].

Therefore, the aim of this study was twofold: (1) to compare objectively measured PA levels of preschool children during time spent in kindergarten versus at home, and (2) to examine associations between selected socio-demographic factors and children's PA levels. This study represents the first objective assessment of PA across both settings in a Polish preschool population.

## Materials and methods

The study was approved by the Bioethics Committee of the University of Rzeszów (no. 2018/01/05) and conducted in accordance with the Declaration of Helsinki. Prior to the commencement of the study, written informed consent was obtained from the parents of the participating children.

### Procedures

The study procedures were conducted in accordance with previously established methodological protocols [43,44]. The required sample size for a cross-sectional study was calculated using a 95% confidence level and a 5% margin of error. In 2018, the population of children aged 5–6 years in Rzeszów was approximately 4,300. Assuming maximum variability ($p = 0.50$), the minimum sample size required was 353 children, after applying the finite population correction. The selection of kindergartens was conducted using the STATISTICA program with sampling without replacement. Consent was secured from the principals of 41 kindergarten. The kindergarten staff distributed consent forms, surveys, and detailed guidelines to the parents of all children enrolled in the selected kindergartens. Parents completed the questionnaire collaboratively. In instances where multiple children from the same family participated, each child was assigned a unique identification code, and parents were directed to complete a separate questionnaire for each child. Parents provided written informed consent. The recruitment period lasted from February 1, 2018, to March 1, 2018.

Signed consents and completed questionnaires were collected by the kindergarten staff. Anthropometric measurements were conducted in a designated room within the kindergarten premises. All measurements were taken between 8:00 and 10:00 am.

## Participants

The inclusion criteria for the study were (1) children aged 5–6 years, (2) whose parents or guardians provided written informed consent and child assent prior to data collection and (3) who were enrolled in kindergartens located in Rzeszów. During the data collection phase, 43 participants were excluded from the study for various reasons: severe anxiety related to testing (n = 5; 0.9% − 1 girl and 4 boys), failure to return or complete the questionnaire (n = 20; 3.5% − 15 girls and 5 boys). Accelerometers were only given to children who had not been excluded for previous reasons (n = 540). Following the initial analysis of the raw data, a further 18 participants (10 girls and 8 boys) were excluded due to a lack of valid accelerometer data (i.e., children with less than 500 minutes of wear time per day on more than one analysed day). Thus, 522 children were included in the final analysis.

## Anthropometric measurements

Body height was measured to the nearest 0.1 cm using a portable stadiometer (Tanita HR-200, Tokyo, Japan). The measurement was taken in a vertical position, barefoot. Body weight was assessed to the nearest 0.1 kg using a body composition analyzer (BC-420 MA, Tanita, Tokyo, Japan). The procedures were carried out in accordance with previous methodological arrangements [45].

## Physical activity

Based on information gathered from kindergartens, it was established that most parents drop their children off at around 8:00 a.m. and collect them before 3 p.m. Initial interviews with parents revealed that most children go to bed at around 8 p.m. Taking this into account, the observation period in kindergartens was set from 8:00 a.m. to 2:00 p.m. (six hours). This was done to ensure consistency across all participants and kindergartens, to capture the period of the day when structured activities typically occur, and to enable reliable comparison of the data across the sample. Furthermore, parents were provided with instructions regarding the recommended hours of attendance at kindergarten. It has been recommended that children spend six hours at kindergarten (8 a.m. – 2 p.m.), with an approximate bedtime of 8 p.m.

An ActiGraph GT3X-BT triaxial accelerometer (ActiGraph, Pensacola, FL, USA) was used to measure PA. The ActiGraph GT3X-BT accelerometer has been widely validated and used in pediatric populations, including preschool and early school-aged children. Validation studies have demonstrated acceptable validity of this devices for estimating PA and activity intensity in children aged 5–9 years using indirect calorimetry and energy expenditure as reference methods [46,47].

The accelerometer was worn on the participant's right hip. Participants were instructed to wear the accelerometer for five consecutive days and nights (Monday through Friday), excluding weekends, 24 hours a day. The study was conducted between April and June, during the spring months, when weather conditions generally support outdoor activity.

ActiGraph data were analyzed using Actilife 6.13. (ActiGraph LLC, Pensacola, Florida, USA). Actigraphy data were collected at a sampling rate of 30 Hz and the Sadeh sleep algorithm [48] was used to detect the sleep period. After the nocturnal sleep episode time was excluded, no-wear time was defined as 60 minutes of consecutive zeros, including two minutes of non-zero pauses [49]. Waking wear time and the PA data were collected in 5-second epoch periods. A wearing time of ≥500 min./day was used as the criterion for a valid day [50]. No separate wearing-time criteria were defined for valid kindergarten and home periods. Cut-off points according to Evenson et al. [51] were chosen to determine time spent at MVPA levels (>2296 counts per minute – CPM). The cut-off points were as follows: sedentary lifestyle: 0–100 CPM, light: 101–2295 CPM, moderate: 2296–4011 CPM, vigorous: 4012-∞ CPM.

The cut-off points proposed by Evenson et al. were chosen due to their higher classification accuracy than other cut-off points, their consistency across studies, and their validation in diverse populations and practical settings [52].

No imputation methods or sensitivity analyses were used for any missing accelerometer data. The statistical analysis included the average time spent in light, moderate, and vigorous PA, as well as sedentary time, across five analysed weekdays. A total of 404,383.7 minutes of waking wear time (averaged across the five analysed days) were analysed, corresponding to an average of 12.9 hours per participant per day. The total waking wear time recorded in kindergarten and at home was 201,230.6 and 202,887.8 minutes, respectively. Less than 1.8% of the total data was excluded due to invalid daily wear time.

The number of steps was also analyzed. Daily step count was calculated as the average daily step count over all valid days. All steps below 1,000 and above 30,000 steps per day were removed and treated as missing data according to the rules of Rowe et al. [53].

## Selected socioeconomic indicators

Parents of children participating in the study were given a questionnaire to complete in order to provide relevant information, including education level and family structure. Selected socioeconomic indicators and socio-demographic characteristics (children's gender and date of birth, place of residence, number of people in the household, parents' education) were self-reported by parents/guardians using a questionnaire. The financial status was assessed via parental self-report using three response options: (1) *Sufficient* – able to meet all basic needs without financial constraints; (2) *Restrictions are necessary* – needing to limit spending to cover essentials; and (3) *Insufficient* – unable to meet basic needs. For analysis, responses (2) and (3) were combined and classified as *Less than sufficient*.

## Statistical analysis

Statistical analysis was performed using SPSS 20 software (IBM, North Harbour, UK). The normal distribution of all quantitative variables was assessed using the Kolmogorov-Smirnov test. Continuous data are presented as the median (Me) and interquartile range (IQR), while categorical variables are presented as the number (n) and percentage (%). To assess significant differences between groups, the Mann–Whitney test was used for independent continuous and dichotomous variables, and the chi-square or exact Fisher tests were used for categorical variables. For parameters with more than two categories, the Kruskal–Wallis test (with chi-square approximation) was used. The Wilcoxon signed-rank test was used to assess significant differences between dependent groups. To assess the effect size in continuous variables, the Cohen's r test was employed, and the Cramér's V test was used for categorical variables. The level of statistical significance was set at $p < 0.05$.

## Results

Table 1 presents the general characteristics of the study population, which comprised 522 participants aged 5–6 years (mean age: 5.5 years; 51.9% female). Statistically significant differences were observed in the place of residence and maternal education level, with a higher proportion of boys living in cities and having mothers with vocational education.

Table 2 presents a comparative analysis of children's PA and sedentary behavior in kindergarten and home settings, stratified by sex. Data are reported as medians with interquartile ranges (IQR). Children exhibited significantly more light PA in kindergarten than at home in the total sample ($p < 0.001$), with a similar trend observed in boys ($p < 0.001$). In contrast, moderate, vigorous, and total MVPA were significantly higher at home than in kindergarten across the total sample, girls, and boys (all $p < 0.001$). Additionally, boys were significantly more sedentary at home compared to kindergarten ($p = 0.040$), and girls took significantly more steps per day at home than in kindergarten ($p = 0.006$). Between-sex differences were also observed, with boys demonstrating higher levels of light, moderate, vigorous, and MVPA than girls in both kindergarten and home settings, while girls accumulated more steps per day ($p < 0.05$).

**Table 1. General characteristics of the study population.**

| | Total sample | Girls | Boys | Significance/ effect size |
|---|---|---|---|---|
| **Sex†** | 522 (100) | 271 (51.9) | 251 (48.1) | p=0.381; V=0.038 |
| **Age (years)*** | 5.5 (1.0) | 6.0 (1.0) | 5.0 (1.0) | p=0.189; r=0.057 |
| **Height (cm) *** | 116.3 (8.6) | 116.0 (8.5) | 116.9 (8.7) | p=0.067; r=0.083 |
| **Body weight (kg) *** | 20.8 (4.6) | 20.4 (4.7) | 21.0 (4.5) | p=0.330; r=0.043 |
| **BMI (kg/m²) *** | 15.3 (1.9) | 15.3 (2.2) | 15.2 (1.7) | p=0.344; r=0.041 |
| **Place of residence†** | | | | |
| **City** | 500 (95.8) | 255 (94.1) | 245 (97.6) | **p=0.046; V=0.087** |
| **Village** | 22 (4.2) | 16 (5.9) | 6 (2.4) | |
| **Education – mother†** | | | | |
| **Primary education** | 2 (0.4) | 1 (0.4) | 1 (0.4) | **p=0.026; V=0.123** |
| **Vocational education** | 16 (3.1) | 3 (1.1) | 13 (5.2) | |
| **Secondary education** | 99 (19.0) | 56 (20.7) | 43 (17.1) | |
| **Higher education** | 405 (77.6) | 211 (77.9) | 194 (77.3) | |
| **Education – father†** | | | | |
| **Primary education** | 5 (1.0) | 3 (1.1) | 2 (0.8) | p=0.936; V=0.029 |
| **Vocational education** | 53 (10.2) | 27 (10.0) | 26 (10.4) | |
| **Secondary education** | 165 (31.6) | 83 (30.6) | 82 (32.7) | |
| **Higher education** | 299 (57.3) | 158 (58.3) | 141 (56.2) | |
| **Number of people living in the household†** | | | | |
| **1** | 5 (1.0) | 4 (1.5) | 1 (0.4) | p=0.666; V=0.081 |
| **2** | 18 (3.4) | 11 (4.1) | 7 (2.8) | |
| **3** | 135 (25.9) | 68 (25.1) | 67 (26.7) | |
| **4** | 258 (49.4) | 129 (47.6) | 129 (51.4) | |
| **5** | 77 (14.8) | 42 (15.5) | 35 (13.9) | |
| **More than 5** | 29 (5.6) | 17 (6.3) | 12 (4.8) | |
| **Financial status†** | | | | |
| **Sufficient** | 483 (92.5) | 253 (93.4) | 230 (91.6) | p=0.239; V=0.080 |
| **Restrictions are necessary** | 36 (6.9) | 18 (6.6) | 18 (7.2) | |
| **Insufficient** | 3 (0.6) | 0 (0.0) | 3 (1.2) | |

Data are presented as: * Median (IQR – Interquartile Range) and † n (%), p – Mann–Whitney U test for continuous variables) or chi-square test for categorical variables, r - Cohen's r effect size test, V - Cramér's V effect size test, significant differences are highlighted in bold, BMI – Body Mass Index.

Table 3 presents the percentage of PA indicators for children in kindergarten and at home. Significant differences were found in the distribution of several activity types. Children engaged in a higher percentage of light PA in kindergarten compared to home, both in the total sample and among boys (both p<0.001). In contrast, the percentage of time spent in moderate PA, vigorous PA, and MVPA was significantly higher at home across the total sample, girls, and boys (all p<0.001). Additionally, boys spent a significantly greater proportion of time in sedentary behavior at home compared to kindergarten (p=0.039). Between-sex differences were also observed, with boys demonstrating higher proportions of light, moderate, vigorous, and MVPA compared to girls in both kindergarten and home settings, while differences in sedentary behavior were less consistent.

Table 4 presents the associations between maternal education and children's PA indicators. Children whose mothers had primary, vocational, or secondary education engaged in significantly more moderate PA (p=0.007), MVPA (p=0.044), and accumulated more steps per day in kindergarten (p=0.042) compared to those whose mothers had higher education.

**Table 2. Levels of physical activity in kindergarten and at home, including between-sex comparisons.**

| Variable | Total sample | | | Girls | | | Boys | | | Girls vs Boys (Kindergarten) | Girls vs Boys (Home) |
|---|---|---|---|---|---|---|---|---|---|---|---|
| | Kinder-garten | Home | Signifi-cance/ effect size | Kinder-garten | Home | Signifi-cance/ effect size | Kinder-garten | Home | Signifi-cance/ effect size | Significance/ effect size | Significance/ effect size |
| Seden-tary | 139.3 (62.3) | 142.0 (72.3) | p=0.071; r=0.079 | 145.0 (62.0) | 144.3 (67.0) | p=0.613; r=0.031 | 131 (67.0) | 136.7 (79.7) | **p=0.040; r=0.130** | **p=0.003; r=0,131** | **p=0.170; r=0.060** |
| Light PA | 233.2 (49.4) | 222.9 (52.7) | **p<0.000; r=0.217** | 227.3 (50.3) | 222.7 (48.7) | p=0.166; r=0.084 | 240.3 (45.0) | 224.0 (57.3) | **p<0.000; r=0.352** | **p<0.000; r=0.172** | **p=0.824; r=0.010** |
| Moder-ate PA | 13.3 (13.0) | 17.6 (14.0) | **p<0.000; r=0.366** | 12.3 (11.0) | 16.0 (13.7) | **p<0.000; r=0.396** | 15.0 (15.2) | 19.5 (16.2) | **p<0.000; r=0.337** | **p=0.005; r=0.124** | **p=0.014; r=0.108** |
| Vigor-ous PA | 3.3 (3.0) | 4.7 (4.8) | **p<0.000; r=0.384** | 3.3 (3.3) | 4.5 (4.8) | **p<0.000; r=0.383** | 3.5 (3.0) | 4.7 (4.7) | **p<0.000; r=0.390** | **p=0.942; r=0.003** | **p=0.555; r=0.026** |
| MVPA | 15.7 (14.0) | 20.6 (18.3) | **p<0.000; r=0.387** | 14.0 (13.0) | 19.0 (16.0) | **p<0.000; r=0.411** | 17.3 (16.0) | 22.7 (21.7) | **p<0.000; r=0.364** | **p=0.003; r=0.131** | **p=0.014; r=0.108** |
| Steps/ day | 3111.0 (2023.3) | 3315.5 (2095.7) | **p=0.012; r=0.110** | 2894.0 (1948.3) | 3267.7 (2044.7) | **p=0.006; r=0.168** | 3266.7 (2298.0) | 3403.7 (2230.7) | p=0.414; r=0.052 | **p=0.003; r=0.132** | p=0405; r=0.036 |

Data are presented as: Median (IQR – Interquartile Range), MVPA – moderate to-vigorous physical activity, PA – physical activity, p – Wilcoxon signed-rank test/ Mann-Whitney' test, r - Cohen's r effect size test, significant differences are highlighted in bold.

**Table 3. Percentage of physical activity indicators in kindergarten and at home, including between-sex comparisons.**

| Variable | Total sample | | | Girls | | | Boys | | | Girls vs Boys (Kindergarten) | Girls vs Boys (Home) |
|---|---|---|---|---|---|---|---|---|---|---|---|
| | Kinder-garten | Home | Significance/ effect size | Kinder-garten | Home | Significance/ effect size | Kinder-garten | Home | Significance/ effect size | Significance/ effect size | Significance/ effect size |
| Seden-tary | 33.2 (14.8) | 33.8 (17.2) | p=0.071; r=0.079 | 34.5 (14.8) | 34.4 (16) | p=0.610; r=0.031 | 31.2 (16) | 32.5 (19) | **p=0.039; r=0.130** | **p=0.003; r=0.131** | **p=0.172; r=0.060** |
| Light PA | 55.7 (11.5) | 53.1 (12.5) | **p<0.000; r=0.243** | 54.4 (11.3) | 53.0 (11.6) | p=0.100; r=0.100 | 57.3 (10.4) | 53.3 (13.7) | **p<0.000; r=0.390** | **p<0.000; r=0.170** | **p=0.859; r=0.008** |
| Moder-ate PA | 3.2 (3.1) | 4.2 (3.3) | **p<0.000; r=0.366** | 2.9 (2.6) | 3.8 (3.3) | **p<0.000; r=0.395** | 3.6 (3.6) | 4.6 (3.8) | **p<0.000; r=0.337** | **p=0.005; r=0.124** | **p=0.014; r=0.108** |
| Vigor-ous PA | 0.8 (0.7) | 1.1 (1.2) | **p<0.000; r=0.383** | 0.8 (0.8) | 1.1 (1.2) | **p<0.000; r=0.386** | 0.8 (0.7) | 1.1 (1.1) | **p<0.000; r=0.387** | **p=0.843; r=0.009** | **p=0.555; r=0.026** |
| MVPA | 3.7 (3.3) | 4.9 (4.4) | **p<0.000; r=0.386** | 3.3 (3.1) | 4.5 (3.8) | **p<0.000; r=0.407** | 4.1 (3.8) | 5.4 (5.2) | **p<0.000; r=0.365** | **p=0.003; r=0.130** | **p=0.013; r=0.108** |

Data are presented as: Median (IQR – Interquartile Range), MVPA – moderate to-vigorous physical activity, PA – physical activity, p – Wilcoxon signed-rank test, r - Cohen's r effect size test, significant differences are highlighted in bold.

Table 5 presents the associations between paternal education and PA indicators in children. Significantly higher levels of light PA (p = 0.023), MVPA (p = 0.024), and steps/day (p = 0.026) in kindergarten were observed among children whose fathers had primary, vocational, or secondary education compared to those with higher-educated fathers.

Table 6 presents the associations between financial conditions in the family and children's PA indicators. A significantly higher level of MVPA in kindergarten was observed among children from families with less than sufficient financial resources compared to those with sufficient conditions (p = 0.038).

**Table 4. Associations between mother's education and physical activity indicators.**

| Variable | Mother's education | | |
|---|---|---|---|
| | Primary/ vocational/ secondary education | Higher education | p |
| Sedentary (min.) in kindergarten | 133.7 (57.7) | 140.3 (64.0) | p=0.500; r=0.030 |
| Sedentary (min.) at home | 140.7 (63.7) | 143 (73.7) | p=0.693; r=0.017 |
| Light PA (min.) in kindergarten | 236.5 (39.0) | 231.7 (53.0) | p=0.252; r=0.050 |
| Light PA (min.) at home | 223.0 (52.0) | 222.8 (52.8) | p=0.557; r=0.026 |
| Moderate PA (min.) in kindergarten | 15.7 (13.7) | 12.5 (12.7) | **p=0.007; r=0.118** |
| Moderate PA (min.) at home | 18.3 (13.7) | 17.3 (14.3) | p=0.726; r=0.015 |
| Vigorous PA (min.) in kindergarten | 3.0 (3.8) | 3.3 (3.0) | p=0.555; r=0.026 |
| Vigorous PA (min.) at home | 4.5 (5.0) | 4.7 (4.8) | p=0.777; r=0.012 |
| MVPA (min.) in kindergarten | 17.0 (17.8) | 15.0 (13.7) | **p=0.044; r=0.088** |
| MVPA (min.) at home | 21.7 (18.0) | 20.3 (18.5) | p=0.277; r=0.048 |
| Steps in kindergarten | 3291.3 (2146.3) | 3012.3 (2040.3) | **p=0.042; r=0.089** |
| Steps at home | 3510.0 (1918.7) | 3267.7 (2118.7) | p=0.530; r=0.027 |

Data are presented as: Median (IQR – Interquartile Range), MVPA – moderate to-vigorous physical activity, PA – physical activity, p – Mann–Whitney U test, r - Cohen's r effect size test, significant differences are highlighted in bold.

**Table 5. Associations between father's education and physical activity indicators.**

| Variable | Father's education | | |
|---|---|---|---|
| | Primary/ vocational/ secondary education | Higher education | p |
| Sedentary (min.) in kindergarten | 136.0 (67.0) | 140.1 (62.5) | p=0.222; r=0.053 |
| Sedentary (min.) at home | 140.0 (73.3) | 143.2 (72.8) | p=0.770; r=0.013 |
| Light PA (min.) in kindergarten | 236.6 (45.3) | 229.8 (51.4) | **p=0.023; r=0.099** |
| Light PA (min.) at home | 224.0 (52.7) | 222.6 (52.4) | p=0.605; r=0.023 |
| Moderate PA (min.) in kindergarten | 13.7 (13.7) | 12.7 (12.9) | p=0.124; r=0.067 |
| Moderate PA (min.) at home | 15.9 (13.7) | 18.5 (14.3) | p=0.544; r=0.027 |
| Vigorous PA (min.) in kindergarten | 3.3 (3.7) | 3.3 (2.9) | p=0.283; r=0.047 |
| Vigorous PA (min.) at home | 4.4 (5.2) | 5.0 (4.7) | p=0.692; r=0.017 |
| MVPA (min.) in kindergarten | 16.7 (16.0) | 14.8 (13.0) | **p=0.024; r=0.099** |
| MVPA (min.) at home | 20.4 (17.0) | 20.7 (20.8) | p=0.545; r=0.027 |
| Steps in kindergarten | 3208.3 (1947.7) | 2905.5 (2215.2) | **p=0.026; r=0.097** |
| Steps at home | 3369.8 (2166.3) | 3302.4 (2056) | p=0.792; r=0.012 |

Data are presented as: Median (IQR – Interquartile Range), MVPA – moderate to-vigorous physical activity, PA – physical activity, p – Mann–Whitney U test, r - Cohen's r effect size test, significant differences are highlighted in bold.

## Discussion

To our knowledge this is the first study among Polish preschool children comparing individual levels of objectively assessed PA during their time in kindergarten versus time spent at home. Our study showed that Polish preschool children were involved in MVPA during 3.7% of time spent in kindergarten and 4.9% of time spent after that – at home.

At home, boys exhibited significantly higher PA parameters, with median MVPA values of 22.7 minutes, compared to 19 minutes for girls. Similarly, in kindergarten, boys again demonstrated significantly higher PA parameters, with median MVPA values of 17.3 min for boys and 14 minutes for girls. These findings align with existing literature indicating that preschool boys tend to be more active than their female counterparts [54,55]. In our study, the average MVPA during

**Table 6.** Associations between financial conditions in the family and physical activity indicators.

| Variable | Financial conditions in the family | | |
|---|---|---|---|
| | Sufficient | Less than sufficient | p |
| **Sedentary (min.) in kindergarten** | 140.0 (62.0) | 127.7 (81.7) | p=0.059; r=0.083 |
| **Sedentary (min.) at home** | 142.3 (75.0) | 136.7 (57.7) | p=0.746; r=0.014 |
| **Light PA (min.) in kindergarten** | 233.5 (48.8) | 230.3 (47.5) | p=0.241; r=0.051 |
| **Light PA (min.) at home** | 222.0 (53.3) | 233.0 (42.7) | p=0.159; r=0.062 |
| **Moderate PA (min.) in kindergarten** | 13.3 (13.0) | 15.7 (20.0) | p=0.140; r=0.065 |
| **Moderate PA (min.) at home** | 18.0 (14.0) | 15.7 (16.7) | p=0.807; r=0.011 |
| **Vigorous PA (min.) in kindergarten** | 3.3 (3.0) | 3.3 (2.4) | p=0.863; r=0.008 |
| **Vigorous PA (min.) at home** | 4.7 (4.7) | 4.7 (5.3) | p=0.858; r=0.008 |
| **MVPA (min.) in kindergarten** | 15.3 (14.0) | 18.3 (16.0) | **p=0.038; r=0.091** |
| **MVPA (min.) at home** | 20.3 (18.5) | 24.7 (17.3) | p=0.317; r=0.044 |
| **Steps in kindergarten** | 3070.7 (2027.0) | 3390.3 (2401.3) | p=0.317; r=0.044 |
| **Steps at home** | 3286.7 (2123.3) | 3493.7 (1891.3) | p=0.248; r=0.051 |

Data are presented as: Median (IQR – Interquartile Range), MVPA – moderate to-vigorous physical activity, PA – physical activity, p – Mann–Whitney U test, r - Cohen's r effect size test, significant differences are highlighted in bold.

kindergarten attendance was 15.7 minutes, which is notably higher than the 11.45 minutes reported by Vanderloo et al. [56] for children in the United States. Although formal interaction effects between sex and setting (home vs. kindergarten) were not examined, the observed differences in MVPA between boys and girls were of comparable magnitude in both environments. Boys accumulated approximately 3–4 more minutes of MVPA than girls both at home and in kindergarten, suggesting that sex-related differences in PA are relatively stable across settings rather than being amplified in one specific environment.

Despite access to playgrounds, peer interactions, and structured activities, children in our study exhibited higher PA levels at home than in kindergarten. This observation aligns with the conclusions of Foweather et al. [57] and O'Neill et al. [58], who noted that children in kindergarten are generally less active within that environment compared to outside. Conversely, other studies highlight the potential of preschool facilities to enhance PA levels [59,60]. Although kindergartens offer access to peers and play spaces, organisational and structural factors may limit opportunities for higher-intensity PA during the kindergarten day. Previous studies indicate that daily schedules are often dominated by teacher-led and low-intensity activities, while indoor space constraints, safety regulations, and limited outdoor time may further restrict opportunities for vigorous movement [28–30,38,39]. In contrast, the home environment typically allows greater flexibility, fewer behavioural constraints, and more opportunities for child-directed and spontaneous play, which may facilitate engagement in higher-intensity activities [25,61]. These findings suggest that increasing MVPA in kindergarten settings may require adjustments in daily organisation, greater emphasis on active play, and improved use of indoor and outdoor spaces to better support children's natural movement behaviours.

The average number of steps observed in the present study was lower than values reported in several international studies [8,62–64], although comparable to some context-specific analyses [65]. Differences across studies may reflect variation in measurement protocols, accelerometer settings, and cultural or environmental factors influencing daily activity patterns.

While higher parental education and financial resources are often linked to greater access to organised activities and supportive environments [66–69], such associations are not consistently observed in younger populations [70–72]. The present findings suggest that socioeconomic influences on PA may be context-dependent rather than uniform across settings. In particular, kindergarten may serve as an important environment for promoting MVPA among children from less advantaged families, potentially reducing inequalities in early childhood movement behaviours.

Evidence supporting a positive association between parental education and PA is more consistently observed in older children than in preschool populations. Although children of higher-educated parents often demonstrate higher PA levels and lower sedentary behaviour, likely due to greater parental support and structured routines [68,69], this relationship is not consistently confirmed in younger children. For example, Malmo et al. [70] reported no significant association between parental education and leisure-time PA in preschool-aged children. These inconsistencies may reflect developmental differences and cross-country variation in access to structured PA opportunities.

Family financial conditions are typically linked to greater access to supportive environments for PA; however, the present results indicate that children from less affluent families accumulated higher MVPA during kindergarten hours, with no differences observed at home. This suggests that the association between financial resources and PA in early childhood may be context-specific, with kindergarten playing a particularly important role for children from less advantaged families.

### Strengths and limitations

One of the strengths of this study is that it provides objective data on children's PA. Notable strengths of this study include the analysis of PA per hour during the child's time in kindergarten and at home. This research may be helpful in identifying families/children who are at greater risk of promoting an inactive lifestyle among young children. In kindergarten, children may have limited mobility due to all the internal norms, practices and rules that reinforce a sedentary lifestyle (e.g., early teacher-led classes, limited indoor space, or lack of space for PA in the winter) and additionally it is important to remember that parents play an important role in promoting a healthy and active lifestyle for their children.

In addition, our study provided new insights, particularly regarding the association between selected socio-demographic factors and kindergarteners, based on a relatively large sample encompassing numerous kindergartens.

However, several limitations should be acknowledged. The study was conducted in Rzeszów and its surrounding areas, which may limit generalisability to other regions. Moreover, due to its cross-sectional design, causal relationships cannot be inferred, nor can long-term trends in PA be established.

The absence of data on parental occupation represents another limitation, as working patterns may influence time availability, supervision, and opportunities for children's PA. Future research should consider including this variable to provide a more comprehensive understanding of contextual determinants.

Methodological limitations related to accelerometry should also be considered. Although hip-worn triaxial accelerometers provide objective and widely validated estimates of movement intensity, they may underestimate non-vertical or upper-body dominant activities such as cycling, climbing, or static strength-based play, and they do not capture contextual information about the activity performed. Furthermore, the use of a single accelerometer may limit the ability to fully capture young children's complex and multi-dimensional movement patterns; multi-sensor approaches could provide a more comprehensive assessment.

The choice of Evenson cut-points constitutes an additional methodological consideration. While these thresholds are widely validated and frequently used in pediatric populations, different cut-point definitions can produce systematically different estimates of activity intensity. Therefore, absolute MVPA values should be interpreted with caution when comparing results across studies using alternative thresholds.

A further limitation concerns the segmentation of kindergarten and home time. Although recommended hours were provided to parents and typical attendance patterns were considered, no objective verification of actual attendance or bedtime was available. In addition, no separate wear-time validity criteria were defined specifically for kindergarten and home segments.

Finally, potential interaction effects between sex and setting were not formally tested. Future research employing repeated-measures or mixed-model approaches should examine whether environmental context differentially influences PA patterns in boys and girls.

The present findings highlight the need to strengthen PA promotion within kindergarten settings. Lower levels of MVPA during kindergarten hours compared to time spent at home indicate that daily schedules and environments should better

support active and spontaneous play. Practical strategies may include increasing opportunities for unstructured PA, improving access to indoor and outdoor play spaces, and supporting staff in integrating movement into daily routines. Kindergartens may also serve as an important setting for reducing inequalities in PA, particularly for children from less advantaged families.

## Conclusions

This study demonstrated significant differences in preschool children's PA levels between kindergarten and home environments. Although children accumulated more light PA during kindergarten hours, levels of MVPA and daily step counts were higher at home. Boys consistently exhibited higher MVPA than girls across both settings, with differences of similar magnitude in kindergarten and at home.

Associations between selected socioeconomic indicators and PA were observed primarily within the kindergarten setting. Children of parents with lower educational attainment and those from families reporting less favorable financial conditions accumulated higher MVPA during kindergarten hours. These findings suggest that socioeconomic influences on PA in early childhood may be context-dependent and shaped by complex environmental and structural mechanisms rather than single explanatory factors.

Kindergarten may therefore represent an important setting for promoting PA and potentially reducing inequalities in movement behaviours among preschool children. Future research should further explore contextual determinants of PA across different environments and examine longitudinal patterns to better understand causal relationships.

## Author contributions

**Conceptualization:** Jarosław Herbert, Piotr Matłosz, Justyna Wyszyńska.

**Data curation:** Jarosław Herbert, Piotr Matłosz, Wojciech Ratkowski.

**Formal analysis:** Jarosław Herbert, Wojciech Ratkowski, Justyna Wyszyńska.

**Investigation:** Jarosław Herbert, Piotr Matłosz, Justyna Wyszyńska.

**Methodology:** Jarosław Herbert, Piotr Matłosz, Justyna Wyszyńska.

**Writing – original draft:** Jarosław Herbert, Piotr Matłosz, Justyna Wyszyńska.

**Writing – review & editing:** Jarosław Herbert, Piotr Matłosz, Wojciech Ratkowski, Justyna Wyszyńska.

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
