## [Decision Letter · Decision Letter 0]

29 Dec 2025

PONE-D-25-56543A comparative analysis of objectively assessed physical activity levels in kindergarten and home among children aged 5 to 6PLOS One

Dear Dr. Herbert,

Thank you for submitting your manuscript to PLOS ONE. After careful consideration, we feel that it has merit but does not fully meet PLOS ONE’s publication criteria as it currently stands. Therefore, we invite you to submit a revised version of the manuscript that addresses the points raised during the review process.

We look forward to receiving your revised manuscript.

Kind regards,

Francesca D'Elia, Ph.D.

Academic Editor

PLOS One

Journal Requirements:

Additional Editor Comments:

Dear Authors,

thank you for submitting your manuscript “A comparative analysis of objectively assessed physical activity levels in kindergarten and home among children aged 5 to 6.”

Both reviewers recognize the relevance of the topic and the potential contribution of your work. However, they also identify several substantial issues that must be addressed before the manuscript can proceed.

Based on their evaluations, I am requesting Major Revisions. Please address the following points in detail:

1. Introduction

The Introduction needs clearer structure and stronger conceptual grounding.

Please ensure that each paragraph has a clear thesis statement, avoid combining unrelated ideas in the same sentence, and integrate the international MVPA literature currently placed in the Discussion.

Both reviewers also request the inclusion of recent systematic reviews on accelerometry challenges in young children and on how kindergarten environments influence PA.

The section on Polish research should be merged and rewritten for clarity.

2. Methods

The Methods section lacks essential detail for transparency and reproducibility.

Please clarify how kindergarten vs. home time segments were defined and validated; provide separate validity criteria for each setting; report excluded data (overall, by setting, by sex); and specify whether imputation was used.

A full description of ActiLife 6.13 processing steps is required, along with justification of Evenson cut-points and discussion of limitations of triaxial accelerometers for non-vertical movements.

Add a sample size calculation and references validating the GT3X-BT for this age group.

3. Results

Please indicate the statistical tests used in each table, report non-parametric results consistently using median/IQR, and include appropriate effect sizes.

4. Discussion

The Discussion should be refocused.

Move background literature to the Introduction, discuss potential sex setting interactions, expand interpretation of higher PA at home, and elaborate on socioeconomic mechanisms.

Add implications for policy and practice and strengthen the Limitations section (cut-points, non-vertical movement, single device, segmentation accuracy).

5. Readability and Conclusions

Ensure consistent use of abbreviations and add line numbering.

Revise the Conclusions to avoid simplistic interpretations of parental education and consider broader socioeconomic explanations.

Please submit a revised manuscript and a detailed point-by-point response.

Sincerely,

Reviewers' comments:

Reviewer's Responses to Questions

**Comments to the Author**

1. Is the manuscript technically sound, and do the data support the conclusions?

Reviewer #1: Yes

Reviewer #2: Partly

2. Has the statistical analysis been performed appropriately and rigorously? 

Reviewer #1: Yes

Reviewer #2: Yes

3. Have the authors made all data underlying the findings in their manuscript fully available?

Reviewer #1: No

Reviewer #2: Yes

4. Is the manuscript presented in an intelligible fashion and written in standard English?

Reviewer #1: Yes

Reviewer #2: Yes

5. Review Comments to the Author

Reviewer #1: Introduction p2

In this section there are numerous compound sentences where two ideas are joined with “and”. A compound sentence should not address 2 disparate ideas. Most paragraphs lack a thesis statement so then the paragraph is a collection of ideas instead of presenting a cohesive idea.

Re-phrase the opening sentence to begin with something other than “there”. One way to write this sentence would be: Adequate levels of physical activity for school-age and preschool children improves……

The second paragraph beginning on p2 does not have a thesis statement. Multiple areas of focus are presented in the paragraph. The paragraph is not cohesive.

P3 – The paragraph beginning with “The level of PA” is not a paragraph because it only contains 2 sentences.

“A sedentary lifestyle” is not a paragraph.

“Over the past two decades, research in Poland has predominantly relied on subjective assessments of PA, employing diagnostic survey methods [36].” – poor sentence structure. A better option would be “Over the past two decades in Poland, multiple research projects utilizing subjective assessments of PA have been conducted.”

Combine these two paragraphs and use a thesis statement. I would recommend introducing this earlier into the Introduction since you are correlating

Tables – add in the type of statistical analysis to help the reader better understand how you arrived at the conclusions.

Results – add in the type of statistical analysis to help the reader better understand the results.

Discussion

This needs to be in your Introduction “A meta-analysis combining accelerometer-measured MVPA estimates from 29 studies involving over 6,000 preschool-age children found that kindergarteners spend only about 5.5% of their time on MVPA per day [40]. Similarly, in subsequent studies, Van Cauwenberghe et al. [41] on a sample of 1004 Australian kindergarteners showed that the MVPA rate (hour-by-hour 8 percentage analysis) averaged 4% in kindergarten and 5% at home. Brown et al. [42] reported that children spent 3% on MVPA in kindergarten. Slightly better outcomes were observed by Pate et al. [43], they used Actigraph for two weeks and showed that kindergarteners spent 13% on MVPA. In one recent study by Kang et al. [44], this was an average of 11.7% per day.”

Synthesize the data presented in the above paragraph instead of presenting each study separately.

Overall in the discussion there is a large amount of information presented that would have been meaningful incorporated into the Introduction. This information from references would make a great Introduction to the issues.

What are the limitations of using accelerometer data for young children? And the use of a single accelerometer instead of using multiples? Might there be limitations with the use of a single accelerometer? Add this information to your limitations

Conclusion

You present that lower education levels are associated with the lower physical activity. Could the real impact be that lower education parents work at unskilled or low skilled work and therefore are physically exhausted when they arrive home from work. Or is the implication that low education parents don’t know that physical activity is important?

Reviewer #2: General comments

I commend the authors and appreciate the opportunity to review their work on “A comparative analysis of objectively assessed physical activity levels in kindergarten and home among children aged 5 to 6“. Overall, the work is promising, but I recommend addressing the following major and minor points before it can be considered further for publication.

Introduction

This section needs strengthening:

•The topic addressed is politically and socially relevant, and I believe the underlying idea is valid. However, the article reads more like a description than a true scientific analysis guided by a clearly defined theoretical framework.

•I recommend that you cite more recent systematic reviews that establish the consensus on the difficulty of measuring PA in this age group (the 'accelerometry problem'), thus justifying the study's design choices. Consider for instance detailing how kindergarten policy (e.g., space, teacher training) influences PA levels (e.g., studies showing small indoor spaces constrain activity).

Data Collection

The study compares "kindergarten" vs. "home" PA. How were the exact start and end times for each setting confirmed? How was the accuracy of the segmentation procedure verified? Was this based purely on parent logs/self-report or an objective method?

Physical activity data processing

As it is, the information in this subsection is too scarce and requires detailed clarification to ensure the rigor, comparability and reproducibility of the findings.

It is stated that a wearing time of ≥500 min./day was used as the criterion for a valid day:

•It is important to clearly define the separate criteria for a "valid kindergarten day" and a "valid home day" (in minutes/hours).

•Quantify the impact of missing data by indicating the percentage of raw data excluded (if any) from the total recorded minutes/hours due to invalid wear time or other factors.

•Stratify the percentage of missing/excluded raw data by both setting (kindergarten/home) and sex.

• State whether imputation methods or sensitivity analyses were used for any missing accelerometer bouts or days.

•How much data were actually used for the final analyses? This should be clearly specified and thoroughly described.

•Could you detail the all the steps that were followed when analysing ActiGraph data using Actilife 6.13 to ensure reproducibility?

•Justify the choice of Evenson cut-points over others validated for this age group, acknowledging the known variability of estimates.

•The ActiGraph GT3X-BT is a triaxial accelerometer, meaning it measures acceleration along three orthogonal axes including the Vertical Axis. VA often fails to capture activities like climbing or floor play effectively. How did you address the limitation that non-vertical movement (common in preschool play) might have been underestimated.

Sample: Report sample size calculation or power analysis used.

ActiGraph GT3X-BT accelerometer: Has this device been validated for use in children aged 5 to 6? Has it been used in previous studies? Please provide consistent references supporting the use of this device in similar populations.

Results: Since non-parametric tests were used, ensure all results are consistently reported using median/IQR and that the effect size (e.g., rank correlation or rank difference) is reported alongside p-values, to demonstrate the magnitude of the observed SSI associations.

Discussion

•Your findings show that boys have consistently higher PA levels in both environments. This aligns with extensive similar work, which shows that boys typically engage in activities that generate higher MVPA density than girls' activities - Did the analysis account for potential interaction effects between setting (home vs. kindergarten) and sex? For instance, is the difference between boys and girls more pronounced in one setting?

•Discuss the practical implications of the finding that home PA is higher. Based on similar work, what are the likely practical gaps in the kindergarten that lead to this deficit?

•The finding that children from families with higher SSI (sufficient financial conditions) tend to be more physically active, as implied in Table 6, is consistent with similar work linking higher SSI to greater access to movement-stimulating toys, safe spaces, and parental support. This connection should be elaborated.

•Add a section implication for policy and practice

Limitations: It is important to acknowledge the reliance on the Evenson cut-points as a key study limitation, noting that the reported absolute MVPA values may be systematically different from studies using other validated cut-points.

Readability

•Be consistent in using already defined abbreviations from early paragraphs to latter sections e.g. physical activity as PA……. Use PA in succeeding sections consistently after defining it in paragraph 1 of the introduction. Check paragraph 7 of the introduction as an example of this inconsistency.

•Line numbering is key in enhancing readability and ease of reference. This is missing.

Conclusions supported by data

•The conclusions are supported, but the robustness is questioned by the technical details regarding accelerometer data processing and reporting of effect sizes.

6. PLOS authors have the option to publish the peer review history of their article (what does this mean?). If published, this will include your full peer review and any attached files.

Reviewer #1: No

Reviewer #2: **Yes:**Stanley Kagunda Kinuthia

---

## [Author Response · Author response to Decision Letter 1]

23 Feb 2026

Responses to the Reviewers

1. Introduction The Introduction needs clearer structure and stronger conceptual grounding. Please ensure that each paragraph has a clear thesis statement, avoid combining unrelated ideas in the same sentence, and integrate the international MVPA literature currently placed in the Discussion. Both reviewers also request the inclusion of recent systematic reviews on accelerometry challenges in young children and on how kindergarten environments influence PA. The section on Polish research should be merged and rewritten for clarity.

Response: Thank you for this comment. The Introduction has been thoroughly revised to improve its structure and conceptual clarity. Each paragraph now has a clear thesis statement and focuses on a single thematic issue. We also added recent systematic reviews addressing methodological challenges of accelerometry in young children and evidence on how kindergarten environments influence physical activity. In addition, the section on Polish research has been merged and rewritten to improve clarity and coherence.

2. Methods The Methods section lacks essential detail for transparency and reproducibility. Please clarify how kindergarten vs. home time segments were defined and validated; provide separate validity criteria for each setting; report excluded data (overall, by setting, by sex); and specify whether imputation was used. A full description of ActiLife 6.13 processing steps is required, along with justification of Evenson cut-points and discussion of limitations of triaxial accelerometers for non-vertical movements. Add a sample size calculation and references validating the GT3X-BT for this age group.

Response: We sincerely thank the Reviewer for these valuable suggestions. The Methods section has been thoroughly revised to improve transparency and reproducibility, including clarification of kindergarten and home time segmentation, wear-time criteria, excluded data, absence of imputation, detailed ActiLife 6.13 processing procedures, and justification of the Evenson cut-points. A formal sample size calculation, validation references for the GT3X-BT in this age group, and a discussion of accelerometer limitations have also been added.

3. Results Please indicate the statistical tests used in each table, report non-parametric results consistently using median/IQR, and include appropriate effect sizes.

Response: Thank you for this comment. The Results section and all tables have been revised to clearly indicate the statistical tests used, present non-parametric data consistently as median (IQR), and include appropriate effect size measures.

4. Discussion The Discussion should be refocused. Move background literature to the Introduction, discuss potential sex setting interactions, expand interpretation of higher PA at home, and elaborate on socioeconomic mechanisms. Add implications for policy and practice and strengthen the Limitations section (cut-points, non-vertical movement, single device, segmentation accuracy).

Response: Thank you for this insightful comment; the Discussion has been comprehensively revised in line with all suggestions, including structural refocusing, expanded interpretation, added policy implications, and a strengthened Limitations section.

5. Readability and Conclusions Ensure consistent use of abbreviations and add line numbering. Revise the Conclusions to avoid simplistic interpretations of parental education and consider broader socioeconomic explanations. Please submit a revised manuscript and a detailed point-by-point response. Sincerely, Academic Editor, PLOS ONE

Response: Thank you for these helpful suggestions. The manuscript has been revised to improve readability and consistency of abbreviations, line numbering has been added, the Conclusions have been refined, and a detailed point-by-point response is provided.

We would like to sincerely thank the Academic Editor for the careful evaluation of our manuscript and for the constructive and detailed guidance provided throughout the review process. We greatly appreciate the time and expertise invested in improving the quality, clarity, and methodological rigor of our work. We believe that the revisions undertaken in response to these comments have substantially strengthened the manuscript, and we hope that it now meets the standards required for publication in PLOS ONE.

Reviewer #1:

• Introduction p2 In this section there are numerous compound sentences where two ideas are joined with “and”. A compound sentence should not address 2 disparate ideas. Most paragraphs lack a thesis statement so then the paragraph is a collection of ideas instead of presenting a cohesive idea.

• Re-phrase the opening sentence to begin with something other than “there”. One way to write this sentence would be: Adequate levels of physical activity for school-age and preschool children improves……

• The second paragraph beginning on p2 does not have a thesis statement. Multiple areas of focus are presented in the paragraph. The paragraph is not cohesive.

• P3 – The paragraph beginning with “The level of PA” is not a paragraph because it only contains 2 sentences.

• “A sedentary lifestyle” is not a paragraph. “Over the past two decades, research in Poland has predominantly relied on subjective assessments of PA, employing diagnostic survey methods [36].” – poor sentence structure. A better option would be “Over the past two decades in Poland, multiple research projects utilizing subjective assessments of PA have been conducted.”

• Combine these two paragraphs and use a thesis statement. I would recommend introducing this earlier into the Introduction since you are correlating

Response: Thank you for these detailed and constructive comments. In response, the entire Introduction has been rewritten from scratch. The revised version now has a clear and coherent structure, with each paragraph built around a single thesis statement and without compound sentences combining unrelated ideas. Paragraph length and cohesion have been improved, the opening sentence has been rephrased, and the sections on sedentary behavior and Polish research have been merged and rewritten for clarity and flow.

• Tables – add in the type of statistical analysis to help the reader better understand how you arrived at the conclusions.

• Results – add in the type of statistical analysis to help the reader better understand the results.

Response: Thank you for this comment. The type of statistical analyses has now been clearly specified in both the tables and the Results section to improve clarity and transparency.

• Discussion This needs to be in your Introduction “A meta-analysis combining accelerometer-measured MVPA estimates from 29 studies involving over 6,000 preschool-age children found that kindergarteners spend only about 5.5% of their time on MVPA per day [40]. Similarly, in subsequent studies, Van Cauwenberghe et al. [41] on a sample of 1004 Australian kindergarteners showed that the MVPA rate (hour-by-hour 8 percentage analysis) averaged 4% in kindergarten and 5% at home. Brown et al. [42] reported that children spent 3% on MVPA in kindergarten. Slightly better outcomes were observed by Pate et al. [43], they used Actigraph for two weeks and showed that kindergarteners spent 13% on MVPA. In one recent study by Kang et al. [44], this was an average of 11.7% per day.” Synthesize the data presented in the above paragraph instead of presenting each study separately.

Response: Thank you for this comment. The cited findings have been synthesized into a concise summary and moved to the Introduction, replacing the previous study-by-study description.

• Overall in the discussion there is a large amount of information presented that would have been meaningful incorporated into the Introduction. This information from references would make a great Introduction to the issues.

Response: The Discussion section has been revised and streamlined. General background information and extended literature summaries were reduced to avoid overlap with the Introduction. The revised Discussion now focuses more directly on interpretation of the present findings and their contextual implications.

• What are the limitations of using accelerometer data for young children? And the use of a single accelerometer instead of using multiples? Might there be limitations with the use of a single accelerometer? Add this information to your limitations

Response: We addressed this issue in the 'Limitations' subsection.

• Conclusion You present that lower education levels are associated with the lower physical activity. Could the real impact be that lower education parents work at unskilled or low skilled work and therefore are physically exhausted when they arrive home from work. Or is the implication that low education parents don’t know that physical activity is important?

Response: Thank you for this important comment. The Conclusions section has been revised to avoid simplified interpretations of parental education and to reflect the complex and multifactorial nature of socioeconomic influences on children’s physical activity.

Reviewer #2:

General comments I commend the authors and appreciate the opportunity to review their work on “A comparative analysis of objectively assessed physical activity levels in kindergarten and home among children aged 5 to 6“.

Overall, the work is promising, but I recommend addressing the following major and minor points before it can be considered further for publication.

Introduction This section needs strengthening:

• The topic addressed is politically and socially relevant, and I believe the underlying idea is valid. However, the article reads more like a description than a true scientific analysis guided by a clearly defined theoretical framework.

• I recommend that you cite more recent systematic reviews that establish the consensus on the difficulty of measuring PA in this age group (the 'accelerometry problem'), thus justifying the study's design choices.

• Consider for instance detailing how kindergarten policy (e.g., space, teacher training) influences PA levels (e.g., studies showing small indoor spaces constrain activity).

Response: Thank you for these insightful comments. The Introduction has been revised to incorporate a clearer theoretical framework, recent systematic reviews on accelerometry-related measurement challenges, and more explicit discussion of how kindergarten policies and structural factors influence physical activity levels.

• Data Collection The study compares "kindergarten" vs. "home" PA. How were the exact start and end times for each setting confirmed? How was the accuracy of the segmentation procedure verified? Was this based purely on parent logs/self-report or an objective method?

Response: Thank you for these insightful comments. We provide a detailed description of the procedure used in the Methods section. Based on the information gathered from kindergartens, parents were provided with instructions regarding the recommended hours of attendance. It was recommended that children spend six hours at kindergarten (8 a.m. – 2 p.m.), with an approximate bedtime of 8 p.m. As it was impossible to apply an objective method to validate adherence to the suggested hours, this is acknowledged as a potential limitation of the study in the 'Limitations' subsection.

• Physical activity data processing As it is, the information in this subsection is too scarce and requires detailed clarification to ensure the rigor, comparability and reproducibility of the findings. It is stated that a wearing time of ≥500 min./day was used as the criterion for a valid day:

• It is important to clearly define the separate criteria for a "valid kindergarten day" and a "valid home day" (in minutes/hours).

• Response: Thank you for these insightful comments. No separate wearing-time criteria were defined for valid kindergarten and home periods. This information has been added to the Limitations section.

• Quantify the impact of missing data by indicating the percentage of raw data excluded (if any) from the total recorded minutes/hours due to invalid wear time or other factors.

Response: Thank you for these insightful comments. We have included the percentage of data that was excluded due to invalid wear time and other factors.

• Stratify the percentage of missing/excluded raw data by both setting (kindergarten/home) and sex.

Response: Thank you for these insightful comments. We added stratification of excluded participants by sex. However, we cannot stratify the excluded data by setting (kindergarten/home) because separate wearing-time criteria were not defined for valid kindergarten and home periods.

• State whether imputation methods or sensitivity analyses were used for any missing accelerometer bouts or days.

Response: Thank you for these insightful comments. No imputation methods or sensitivity analyses were used for any missing accelerometer data. For the statistical analysis, we averaged the physical activity (PA) data from five analysed weekdays. We added an appropriate sentence to the 'Method' section.

• How much data were actually used for the final analyses? This should be clearly specified and thoroughly described.

Response: Thank you for these insightful comment. We provide a thorough description of the total time included in the statistical analysis.

• Could you detail the all the steps that were followed when analysing ActiGraph data using Actilife 6.13 to ensure reproducibility?

Response: Thank you for these insightful comment. We have now completed the missing steps in our analysis of the ActiGraph data.

• Justify the choice of Evenson cut-points over others validated for this age group, acknowledging the known variability of estimates.

Response: Thank you for these insightful comment. Evenson et al. cut-off points were choose due to higher than others classification accuracy, consistency across studies and validation in diverse populations and practical settings [Philips et al. 2024]. This statement were added in methodology section Phillips SM, Clevenger KA, Bruijns BA, et al. Effect of Accelerometer Cut-Points on Preschoolers’ Physical Activity and Sedentary Time: A Systematic Review and Meta-Analysis. Journal for the Measurement of Physical Behaviour. 2024;7(1): e jmpb.2023-0060. doi:10.1123/jmpb.2023-0060 We added an appropriate justification to the 'Method' section.

• The ActiGraph GT3X-BT is a triaxial accelerometer, meaning it measures acceleration along three orthogonal axes including the Vertical Axis. VA often fails to capture activities like climbing or floor play effectively. How did you address the limitation that non-vertical movement (common in preschool play) might have been underestimated.

Response: Thank you for these insightful comment. We acknowledge that relying on the vertical axis alone may result in an underestimation of the non-vertical and multi-planar movements that are typical of preschool children’s play. To overcome this, age-appropriate and preschool-validated cut-points based on triaxial data were applied. Data were also processed using short epoch lengths of 5 seconds to better reflect young children's intermittent and rapidly changing activity patterns. These methodological choices improve sensitivity to non-vertical movements, such as climbing, crawling and floor-based play. We added this issue to the 'Limitations' section.

• Sample: Report sample size calculation or power analysis used.

Response: Thank you for these insightful comment. A detailed sample size calculation was described in the Methods section.

• ActiGraph GT3X-BT accelerometer: Has this device been validated for use in children aged 5 to 6? Has it been used in previous studies? Please provide consistent references supporting the use of this device in similar populations.

Response: Thank you for these insightful comment. The ActiGraph GT3X-BT accelerometer has been widely validated and used in pediatric populations, including preschool and early school-aged children. Validation studies have demonstrated acceptable validity of this devices for estimating physical activity and acti

---

## [Decision Letter · Decision Letter 1]

1 Apr 2026

PONE-D-25-56543R1A comparative analysis of objectively assessed physical activity levels in kindergarten and home among children aged 5 to 6PLOS One

Dear Dr. Herbert,

Thank you for submitting your manuscript to PLOS ONE. After careful consideration, we feel that it has merit but does not fully meet PLOS ONE’s publication criteria as it currently stands. Therefore, we invite you to submit a revised version of the manuscript that addresses the points raised during the review process.

We look forward to receiving your revised manuscript.

Kind regards,

Francesca D'Elia, Ph.D.

Academic Editor

PLOS One

Journal Requirements:

**Additional Editor Comments:**

Thank you for the revised manuscript.

Both reviewers now recommend minor revisions, but they highlight several points that still require your attention.

Please revise the manuscript accordingly, ensuring that all reviewer comments, both regarding the clarity of the writing and the remaining methodological issues, are fully addressed in the next version.

Reviewers' comments:

Reviewer's Responses to Questions

**Comments to the Author**

1. If the authors have adequately addressed your comments raised in a previous round of review and you feel that this manuscript is now acceptable for publication, you may indicate that here to bypass the “Comments to the Author” section, enter your conflict of interest statement in the “Confidential to Editor” section, and submit your "Accept" recommendation.

Reviewer #1: All comments have been addressed

Reviewer #2: (No Response)

2. Is the manuscript technically sound, and do the data support the conclusions?

Reviewer #1: Yes

Reviewer #2: Partly

3. Has the statistical analysis been performed appropriately and rigorously? 

Reviewer #1: Yes

Reviewer #2: Yes

4. Have the authors made all data underlying the findings in their manuscript fully available?

Reviewer #1: Yes

Reviewer #2: Yes

5. Is the manuscript presented in an intelligible fashion and written in standard English?

Reviewer #1: Yes

Reviewer #2: Yes

6. Review Comments to the Author

Reviewer #1: Within academic writing a paragraph is typically 3 sentences. Inclusion of the very short paragraphs decreases the readability. If the information is crucial re-write a paragraph thesis statement to include the full information and do not chop off pieces into separate blurbs instead of including the information into a paragraph.

Reviewer #2: I still have the following methodological concerns:

1. The authors seem to have fixed some major methodological concerns that I raised by simply listing them as limitations rather than providing more robust analysis or stronger data. The limitations section should not be used as a safety net in methods. For example;

• By failing to provide segmentation accuracy, we miss a solid way to prove exactly when a child was at school versus at home. Consider to either re-analyze the data or provide a more objective validation. While it’s good to be transparent, simply admitting a flaw exists doesn't make the data any more reliable.

• The failure to have separate criteria for a valid kindergarten day and a valid home day means the rigor of the data for each specific setting remains unquantified. Do you still want to qualify/retain this as a limitation?

2.The study continues to report sex (boys vs. girls) and setting (home vs. school) as independent factors without statistically testing how they interact. Are you able to carry out a formal interaction analysis instead of acknowledging this as a study limitation? For instance, do boys get a much bigger boost in activity when they get home compared to girls? Or does the kindergarten environment suppress activity in girls more than it does in boys? The paper has provided the "what," but skipped the statistical test that explains the "how."

3. Without setting-specific wear criteria, it remains unclear if a child's home physical activity is being calculated based on 2 hours or 6 hours of valid data. This raises an issue of data quality.

7. PLOS authors have the option to publish the peer review history of their article (what does this mean?). If published, this will include your full peer review and any attached files.

Reviewer #1: No

Reviewer #2: **Yes:**Kinuthia Stanley

---

## [Author Response · Author response to Decision Letter 2]

28 Apr 2026

I still have the following methodological concerns:

1. The authors seem to have fixed some major methodological concerns that I raised by simply listing them as limitations rather than providing more robust analysis or stronger data. The limitations section should not be used as a safety net in methods. For example;

• By failing to provide segmentation accuracy, we miss a solid way to prove exactly when a child was at school versus at home. Consider to either re-analyze the data or provide a more objective validation. While it’s good to be transparent, simply admitting a flaw exists doesn't make the data any more reliable.

Response:

We thank the Reviewer for this important comment. We fully agree that objective validation of time segmentation between kindergarten and home would strengthen the methodological rigor of the study.

In the present study, parents were instructed to follow standardized daily schedules, with kindergarten attendance occurring between 8:00 a.m. and 2:00 p.m., and home time thereafter until approximately 8:00 p.m. These time frames were consistent across participating institutions and were clearly communicated to parents prior to data collection. Although parents did not maintain detailed logs documenting exact drop-off and pick-up times, they confirmed adherence to these schedules.

We acknowledge that we were unable to objectively verify the exact timing of transitions between settings (e.g., via activity logs or direct observation). Therefore, we are not able to provide a more precise validation of segmentation accuracy or to re-analyze the data using objectively confirmed time stamps. For this reason, this issue has been retained as a methodological limitation.

However, the potential impact of this limitation is mitigated by the high overall data quality. The average daily wear time was approximately 12.9 hours per participant, indicating near-continuous monitoring, and the total recorded wear time was almost equally distributed between kindergarten and home settings. This suggests that both environments were well represented in the dataset. Consequently, although minor misclassification of time segments cannot be excluded, it is unlikely to have materially affected the observed differences between settings.

• The failure to have separate criteria for a valid kindergarten day and a valid home day means the rigor of the data for each specific setting remains unquantified. Do you still want to qualify/retain this as a limitation?

Response: We thank the Reviewer for this comment. We acknowledge that separate wear-time criteria for kindergarten and home periods were not defined, which limits the ability to quantify data rigor at the setting-specific level. As such, we retain this as a methodological limitation.

We would like to clarify that parents did not maintain detailed logs of drop-off and pick-up times. Therefore, even if separate criteria for valid kindergarten and home periods had been defined, it would not have been possible to apply them reliably at the individual level.

Importantly, overall wear time was high (~12.9 hours/day) and evenly distributed between kindergarten and home settings, suggesting balanced data coverage across both environments. Thus, although setting-specific validity thresholds were not applied, the risk of substantial bias due to unequal data representation is likely limited.

2. The study continues to report sex (boys vs. girls) and setting (home vs. school) as independent factors without statistically testing how they interact. Are you able to carry out a formal interaction analysis instead of acknowledging this as a study limitation? For instance, do boys get a much bigger boost in activity when they get home compared to girls? Or does the kindergarten environment suppress activity in girls more than it does in boys? The paper has provided the "what," but skipped the statistical test that explains the "how."

Response: We thank the Reviewer for this valuable comment. In line with this suggestion, we have conducted an additional analysis to formally test the interaction between setting (kindergarten vs. home) and sex. The results of this interaction analysis have now been included in the Results section.

3. Without setting-specific wear criteria, it remains unclear if a child's home physical activity is being calculated based on 2 hours or 6 hours of valid data. This raises an issue of data quality.

Response: We thank the Reviewer for this comment. We acknowledge that, in the absence of setting-specific wear-time criteria, the exact amount of valid data contributing to home and kindergarten periods at the individual level cannot be precisely quantified. This has been recognized as a limitation.

However, we would like to emphasize that overall wear time was high (~12.9 hours/day), indicating near-continuous monitoring, and the total recorded wear time was almost equally distributed between kindergarten and home settings. This suggests that both environments were similarly and adequately represented in the dataset. Therefore, although variability at the individual level cannot be fully excluded, it is unlikely that the results are driven by substantial imbalances in data coverage across settings.

---

## [Editor Report · Decision Letter 2]

3 May 2026

A comparative analysis of objectively assessed physical activity levels in kindergarten and home among children aged 5 to 6

PONE-D-25-56543R2

Dear Dr. Herbert,

We’re pleased to inform you that your manuscript has been judged scientifically suitable for publication and will be formally accepted for publication once it meets all outstanding technical requirements.

Kind regards,

Francesca D'Elia, Ph.D.

Academic Editor

PLOS One

Additional Editor Comments (optional):

Dear Authors,

thank you for submitting the revised version of your manuscript and for providing a clear and comprehensive response to the reviewers’ comments.

I appreciate the substantial effort you have made to address the methodological and analytical issues raised in the previous review rounds. You have strengthened the transparency of the Methods section, clarified key procedural aspects, and incorporated the additional interaction analysis between sex and setting as requested. The Discussion has been improved accordingly, and the limitations are now appropriately acknowledged and contextualized.

While certain methodological constraints, such as the lack of objective validation of time segmentation and the absence of setting‑specific wear‑time criteria, cannot be resolved retrospectively, they are intrinsic to the original dataset and are now fully and transparently reported. These limitations do not compromise the overall validity of the study, which meets PLOS ONE’s standards for methodological soundness and transparency in observational research.

The manuscript is now recommended for acceptance.
---

## [Editor Report · Acceptance letter]

PONE-D-25-56543R2

PLOS One

Dear Dr. Herbert,

I'm pleased to inform you that your manuscript has been deemed suitable for publication in PLOS One. Congratulations! Your manuscript is now being handed over to our production team.

Kind regards,

on behalf of

Dr. Francesca D'Elia

Academic Editor

PLOS One